# Current Challenges of Cold Brew Coffee—Roasting, Extraction, Flavor Profile, Contamination, and Food Safety

**Raven Kwok [1], Kenny Lee Wee Ting [1], Steffen Schwarz [2], Linda Claassen [3,4] and Dirk W. Lachenmeier [3,\*]** 

[1]  Earthlings Coffee Workshop, Soho East, Sublot 16 Ground Floor, Lot 188, Jalan Wan Alwi Lorong 5, Kuching, Sarawak 93350, Malaysia; raven@earthlings-coffee.com (R.K.); kenny@earthlings-coffee.com (K.L.W.T.)
[2]  Coffee Consulate, Hans-Thoma-Strasse 20, 68163 Mannheim, Germany; schwarz@coffee-consulate.com
[3]  Chemisches und Veterinäruntersuchungsamt (CVUA) Karlsruhe, Weissenburger Strasse 3, 76187 Karlsruhe, Germany; lindaclaassen@gmx.de
[4]  Department Life Sciences, University of Applied Sciences Albstadt-Sigmaringen, 72488 Sigmaringen, Germany
\*  Correspondence: Lachenmeier@web.de; Tel.: +49-721-926-5434

**Abstract:** Cold brew coffee has emerged as a new trend over the last decade. However, "cold brew" is an extraction style of ground roasted coffee with water at lower than body temperature (typically 8 °C or room temperature), rather than a beverage per se. Cold brew extraction poses several challenges, including the need for specific optimization depending on the multiple influences of coffee variety and processing, roast degree, grinding, dosage, water composition, turbulence, brew system (drip, immersion etc.), time and temperature, and their interactions. While cold brew is typically characterized by a floral sweetness, over-extraction may lead to abundant acidity and bitterness. To avoid this, an extraction degree of 70% was suggested using shorter time frames (i.e., 2 h at 15 °C with 80 g/L coffee for optimized medium roast profiles). Due to the lack of sterilizing temperatures during preparation, cold brew is significant in the coffee sector because hygiene and food safety requirements pose specific challenges. To avoid microbiological contamination and deterioration in quality, cold brew should be as freshly prepared as possible and shelf-life should be minimized.

**Keywords:** coffee; cold brew; nitro cold brew; roasting; extraction; hygiene; risk assessment; product quality

## 1. Introduction

The history of cold brew coffee can be traced as far back as the 1600s, with a major invention being the Toddy cold-brew coffee system in the 1960s [1]. However, only recently has cold-brew become a growing trend across the entire coffee industry [2]. For example, cold brew sales grew by a remarkable 580% in the US in the period between 2011 and 2016 [3]. Due to its relative novelty, there is an absence of in-depth research into this coffee extraction method [4]. There is also currently a lack of internationally accepted standards or definitions about what cold brew is and under what conditions it is made [5]. The demand for high-quality, cold brew coffee may have been driven by the fact that the segment of iced coffee has been adversely affected in the past "by using old, bitter-tasting, brewed coffee as the base" or "even worse, many have used coffee extracts" [6].

Several cold brew methods such as drip filtration, full immersion, or cold press are available. In general, the term "cold brew" describes a method of preparing a beverage in the form of a certain style of extraction. Cold brew is not necessarily a cold beverage since, unlike iced coffee, cold brew

may be served cold or hot. It is important to differentiate between cold brew and iced coffee, which is a beverage extracted with hot water. Sometimes, mostly in industrial settings, hot brewed beverages are cooled (the so-called "hot bloom" method [5]) and sold as some form of fake cold brew ("called brew").

This article provides an overview about the current knowledge of cold brew coffee preparation, along with practical aspects and potential pitfalls specifically related to hygiene and food safety. Furthermore, open research questions are included, and research challenges are highlighted.

## 2. Materials and Methods

For the first part of the article (Section 3), electronic searches of literature were conducted including the databases PubMed, Google Scholar, and Food Science and Technology Abstracts (FSTA). Search terms used were: ("cold brew" OR "nitro") AND "coffee". The abstracts were screened for relevance regarding aspects of roasting, extraction, flavor profile, contamination, and food safety. Relevant articles were obtained in full text. The searches were complemented by the literature collection of the authors. The major results were discussed using several online training sessions with coffee experts hosted by the professional training centers of Earthlings Coffee Workshop (Malaysia) and Coffee Consulate (Germany), and research plans were derived from this (Section 4). The feedback of the experts is also included in a section about open problems and challenges (Section 5).

## 3. Cold Brew Extraction—What Do We Know?

### 3.1. Roasting for Cold Brew Coffee

Roasting plays a big part in determining what flavors go into the cold brew as it is possible to retain or eliminate various aromas by manipulating different roast levels. Consumer research has shown that the predominant factors influencing cold brew flavor are the roast level [7] and the resting time after roasting [8,9]. In general, the authors suggest avoiding excessive dark roasting, so the resultant coffee avoids unpleasant aromas. Research also found that darker roasts may result in decreased concentrations of compounds [10]. Typical roast profiles for cold brew, filter coffee, and espresso are compared in Figure 1. The biggest variant is with the espresso roast, while cold brew and filter coffee are similar. The cold brew is the fastest roast of the three with a more rapid arrival at phase III. The first phase is more or less the same for all three types. The second, drying phase preserves the acids in the bean. The cold brew roast is slowed down at the very end of its processing to avoid excessive formation of Maillard reaction-based aromas.

### 3.2. Multi-Variate Influences on Cold Brew Extraction

The extraction of cold brew is typically conducted below body temperature, but there is no universally accepted extraction temperature setting. The concept of "cold" is often culturally relative to prevailing natural or artificial environments such as countries with cold winters vs. tropical countries, air conditioning, and refrigeration. The authors, however, believe that temperature settings near or above body temperature move away from the concept of cold brew, and should be critically assessed to avoid misleading food information [11].

Cold brew can be extracted by all the three methods used for hot coffee (see comparison of coffee extraction methods by Angeloni et al. [12]). It can be extracted as (i) cold drip (typically with iced water) in a filtration method, (ii) lixiviation or immersion (grounds in a pot sitting in the water, with or without turbulence), or (iii) cold press (under or overpressure) [11,13,14].

What exactly does the "cold" mean in cold brew? It may be close to 0 °C (i.e., melting ice using the drip method), at fridge temperature (4–8 °C), or at any other temperature below body temperature. There is a much broader range of temperatures than those used for hot coffee extraction. The cold brew extraction time is much longer than for hot brews, and largely depends on the selected temperature. The minimum would be 2 h at 20 °C. It is important to adjust the time/temperature equilibrium to not over or under extract the cold brew.

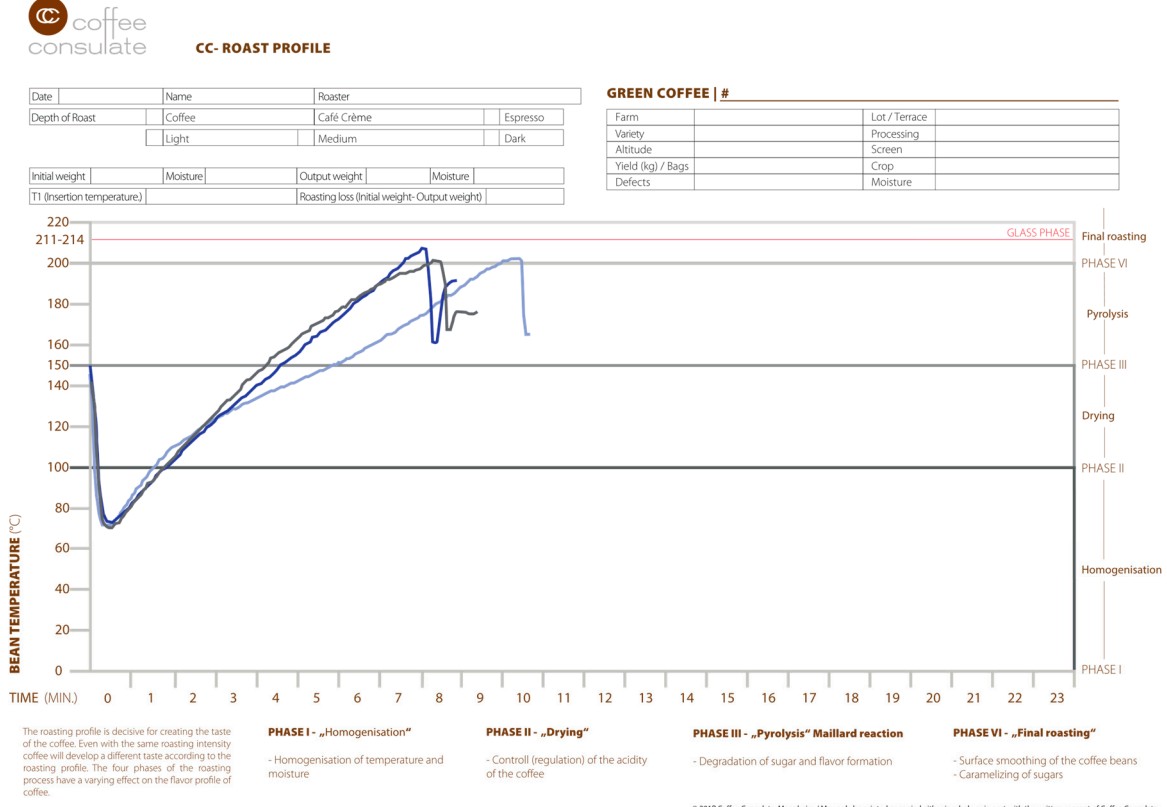

**Figure 1.** Exemplary roasting profiles of cold brew (gray line) compared with filter coffee (dark blue line) and espresso (light blue line) (FZ-94 sample roaster, Coffee-Tech Engineering, Moshav Mazliach, Israel).

The extraction of cold brew coffee depends on several factors including the coffee, roasting, dosage (brew ratio), water temperature, and water composition such as hardness, turbulence (increasing the water contact with the coffee grounds by stirring, agitating, or applying ultrasonic waves), grinding level (particle size and surface, dust), and time (Figure 2). From these parameters, the roasting profile is particularly significant as it influences the acidity that is extracted out of the beans. The grinding surface is also vital because the extraction is influenced if the surface is irregular or rounded.

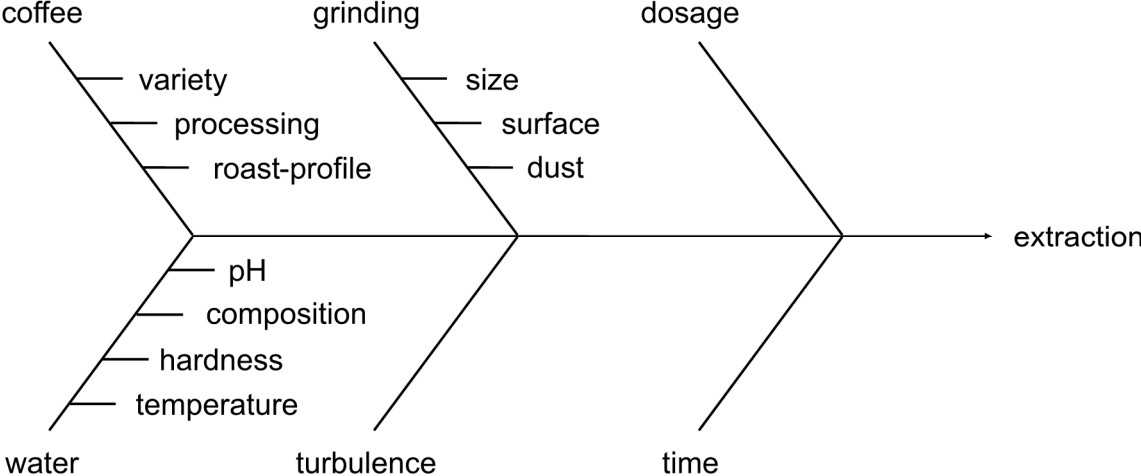

**Figure 2.** Ishikawa diagram of causal influences regarding cold brew coffee extraction.

### 3.3. Hazards during Cold Brew Extraction

Microbiological contamination along with typical chemical contaminants such as pesticides, mycotoxins, acrylamide, and furan must be considered (Figure 3). Cold brew is one the few coffee beverages that includes some microbiological food safety hazards. Risk is usually minimized by the roasting process and fresh, hot brewing. The long brewing time of cold brew can facilitate microbiological activity. Lopez [15] summarized microbiological challenge studies based on personal communications with the industry showing that various species such as *Salmonella*, *Listeria monocytogenes*, and *Escherichia coli* may be viable in cold brew for 7–28 days. Specifically, the potential for *Listeria monocytogenes* to survive in cold brew coffee was judged as being of concern due to its low infective dose [15].

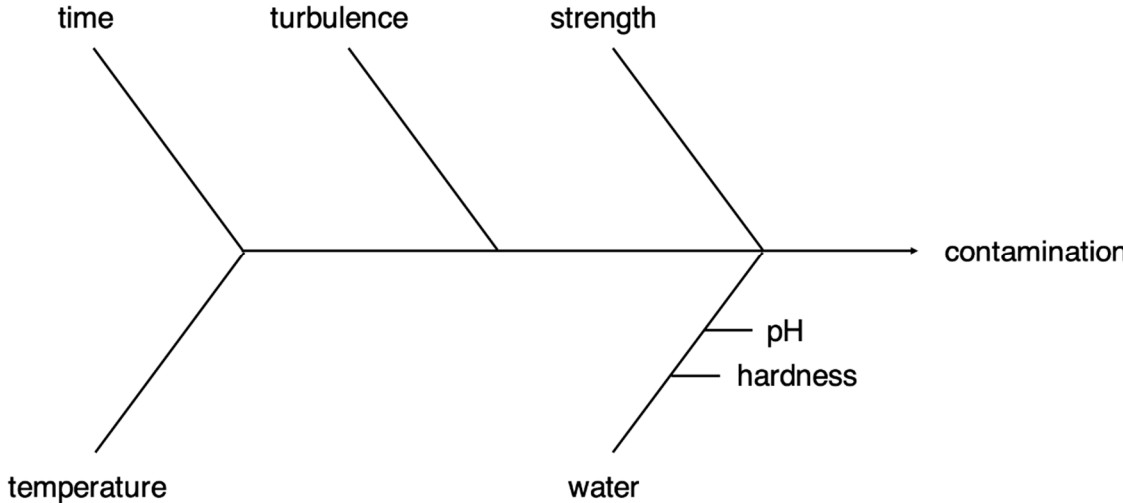

**Figure 3.** Ishikawa diagram of causal influences of extraction on cold brew coffee contamination.

Cold brew producers should be vigilant for any visual or taste changes, which could be an indication of microbial contamination. The risk of microbial growth is strongly related to the extraction temperature and is dramatically reduced at very cold temperatures.

The most common organisms that could spoil cold brew are yeasts (leading to alcoholic fermentation) and lactic or acetic acid bacteria (producing organic acids). For most small-scale producers (e.g., in their coffee shops), it is impossible to work in a fully sterile environment, so the product should have limited storage time. Pathogenic microorganisms such as *Salmonella* or *Listeria* must be avoided.

When filling cans, bottles, or kegs, the use of additives such as ascorbic acid or preservatives may increase microbiological stability and shelf life, as well as heat sterilization or pasteurization (see [5]). However, all these methods are expected to negatively impact the flavor.

Heat-induced coffee contaminants formed during roasting, such as furan or acrylamide [16], were found at similar levels in cold brew when compared with hot brew [17]. While acrylamide may increase with prolonged extraction times, furan may decrease due to volatility [18]. Similarly, other chemical contaminants of coffee such as pesticides or mycotoxins are expected to occur in cold brew when the raw beans are contaminated, but this is not a problem specific to cold brew. Therefore, we believe that microbiological contamination is the exception to the rule that needs specific mitigation measures during the cold brew manufacture.

### 3.4. Extraction Degree

Considering consumer preference in Malaysia and Germany [11,19], 70% extraction (i.e., 70% in relation to the total extractable/soluble amount) may be an optimal starting point for a cold brew coffee recipe to achieve a balanced product (Figure 4). This number does not yet have a scientific basis but

stems from experience considering all aspects of the product, including the risk of microbiological contamination and the flavor. In other words, the optimum extraction results in the best flavor while avoiding contamination. Increased agitation will result in a faster process; cooler temperatures will take longer. Not all the soluble constituents should be extracted, but just enough to get a good taste while minimizing the risk of microbiological contamination. This can also happen in hot extractions such as espresso, where over-extraction will lead to an unpleasant taste [20].

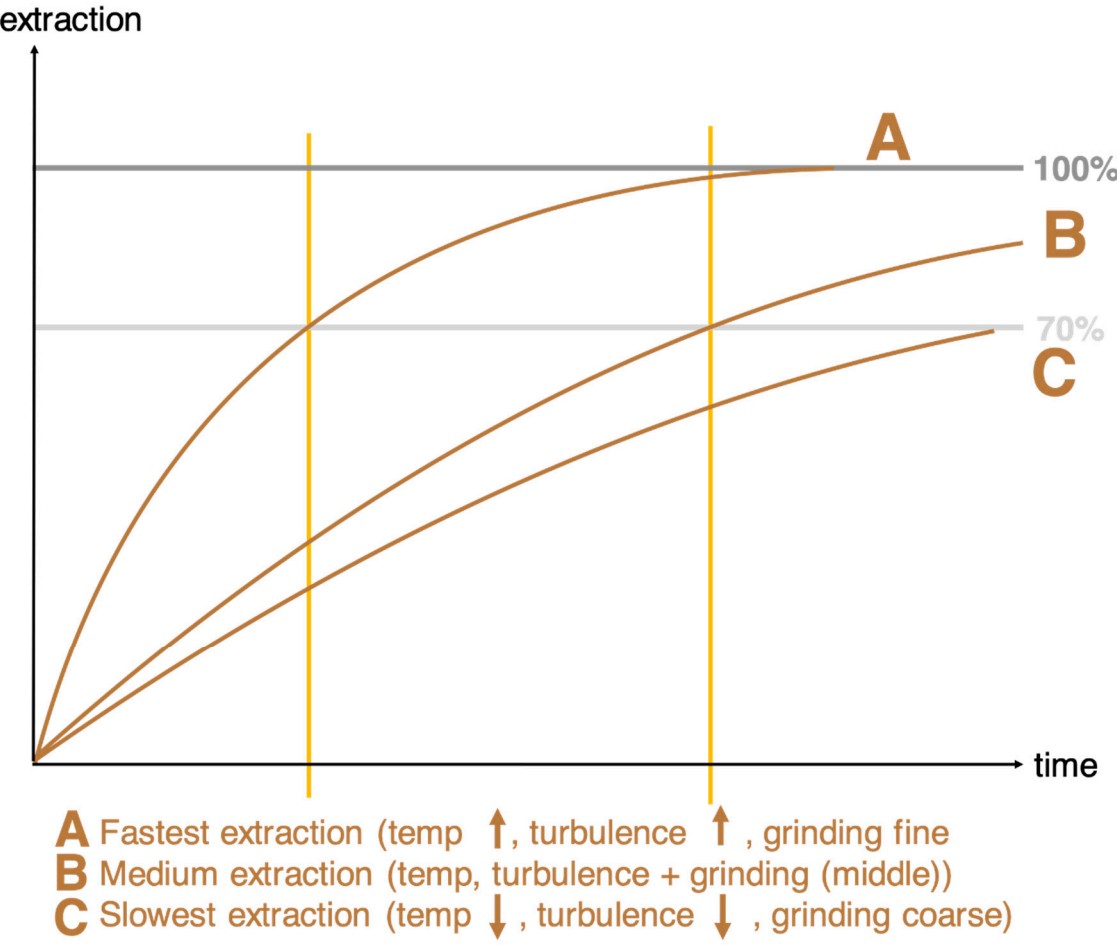

**Figure 4.** Model considerations about cold brew extraction.

Some empirical research confirms the authors' suggestion of 70% extraction. Cordoba et al. [21] reported higher scores in their sensorial evaluation when cold brews were prepared using the shortest time (at 75–86% total dissolved solids compared with a longer extraction time). More research is clearly necessary.

The highest possible extraction, namely 100%, is not desirable in the specialty coffee field. It is better to waste a little bit of coffee, especially the part with the bad flavors. There might be an optimum point, certainly less than 100%, where all the "good" flavors are in and the "bad" flavors are excluded. The equipment and factors mentioned above must all be taken into account to ensure a consistently high-quality product.

### 3.5. Flavor and Taste Profile of Cold Brew Coffee

There are considerable aromatic differences between hot and cold brew coffee [22]. For example, the Coffee Consulate Aroma Wheel [23] shows some grayed out areas for cold brew (Figure 5). Those are the oil-bound aroma groups (i.e., herbs, spices, and nuts), which are suppressed by extraction with cold water. Fruit, floral, and vegetal aromas are more common; also, light roast, and some mineral

and chemical aromas in cases of very dark roasts. Especially in the drip method, with melting water just above 0 °C, the product is characterized by fine, very fruity, and floral aromas. As a variation, some people may extract cold brew with cold milk instead of cold water, which may increase the lipophilic and more polar compounds and flavors. Compounds such as caffeine, which are very water soluble, are typically found in similar levels in cold and hot brew [24].

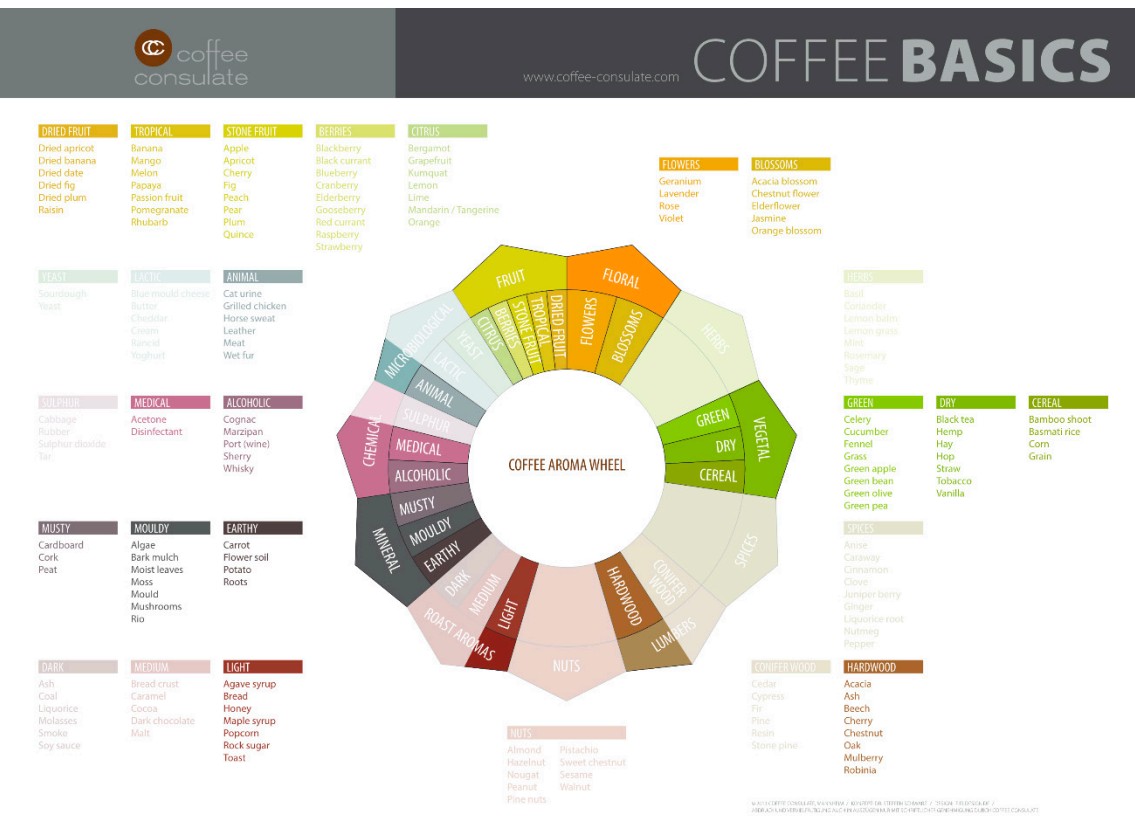

**Figure 5.** Coffee Consulate aroma wheel. Aroma groups that are reduced due to cold brewing are grayed out.

The difference in flavor profile becomes obvious when a cold brew and a standard hot filter coffee extraction of exactly the same coffee variety (oeiras) are compared using the Coffee Consulate flavor profile by a calibrated taste panel (*n* = 3) (Figure 6) [10]. The cold brew profile is more intensive in orange/lemon/cucumber flavors, while filter coffee is more intensive in peach, spice, and microbiological characteristics. Cold brew has slightly less body and is more refreshing.

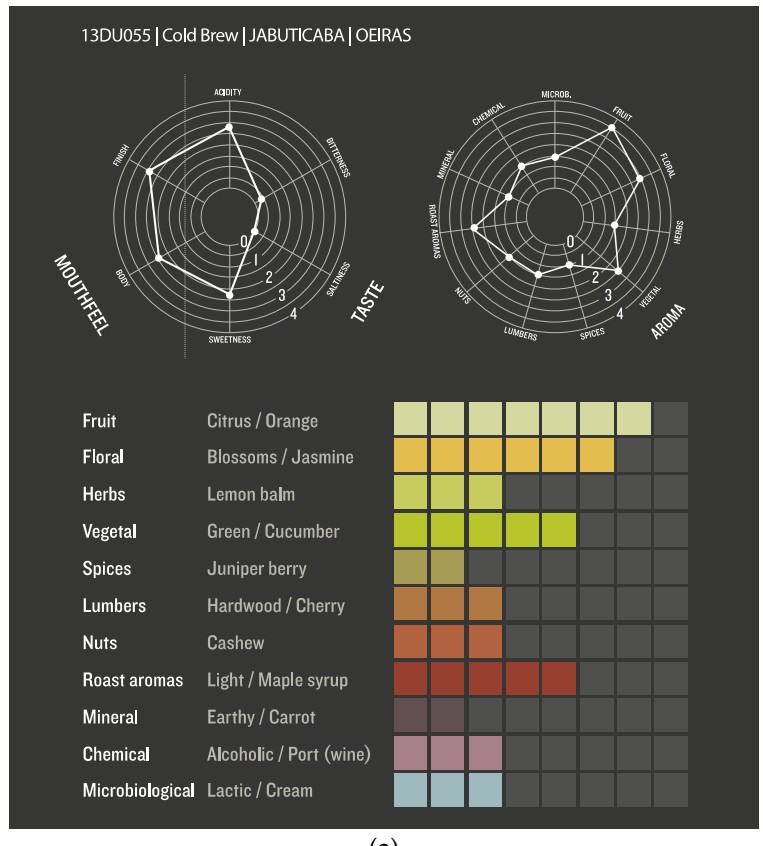

(**a**)

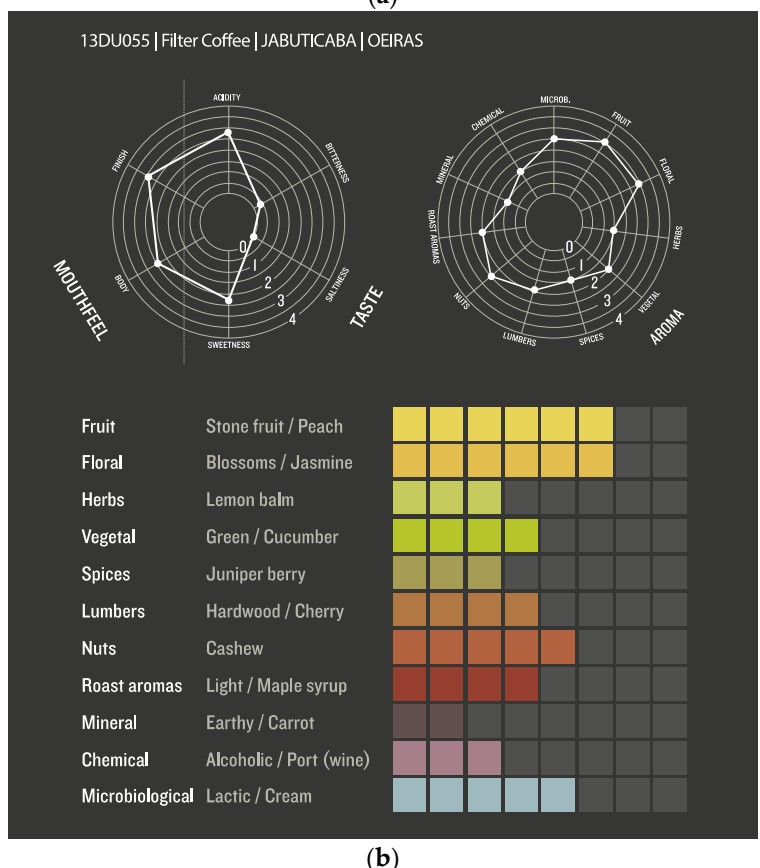

(**b**)

**Figure 6.** Flavor profiles of a cold brew (**a**) compared with a filter coffee (**b**) of the same coffee.

## 4. Cold Brew—Research Plans

Some initial organoleptic experiments with cold brew were initiated during the Intergastra 2020 trade fair (15–19 February 2020, Stuttgart, Germany), which included the Stuttgart Coffee Summit, one of the largest professional coffee trade exhibitions worldwide. The visitors at the Coffee Consulate booth were asked to participate in several ranking order and triangle tests according to ISO 8587 and ISO 4120 methodologies ($n = 60$ for ranking order tests and $n = 25$ for triangle tests). The statistical evaluation of the results is ongoing and will be published later, but some initial trends were that cold brew made with *Coffea arabica* beans was preferred over the one with *Coffea canephora*. There was also a tendency of pulped natural processed Arabica being preferred over fully washed Arabica, potentially due to the higher sweetness of the resulting brew. Another trial showed that consumers significantly preferred cold brew over cooled down hot brew, when the same type of coffee beans were prepared in both instances.

There is not much literature about how cold brew is actually prepared in common practice. Therefore, a questionnaire has been developed [11]. Besides some demographic data and experience with cold brew, the questionnaire investigates the major parameters such as dosage, water composition, brewing temperature, brewing time, grinding degree, coffee variety, roasting degree, and serving styles. The questionnaire was launched during online training in April 2020 and sent to all participants, but also distributed on the Facebook pages of the affiliated institutions and other social network channels. The results are currently under evaluation. However, some initial interesting findings of the first 49 participants as of 22 April 2020, which mostly encompass the participants of several online training sessions, are shortly summarized below.

There was an almost equal distribution between the different cold brew systems (i.e., drip method, commercial systems, French press, mixing in various containers, etc.) with a slight preference (34%) of immersion in containers and filtration afterwards. The applied brew ratios were similarly diverse, with a majority of participants preferring 80–100 g/L. For water quality, soft or medium hard water is preferred. The most preferred extraction temperature is 8 °C followed by 20 °C (see Figure 7). The average brewing time was 16 h (standard deviation 10 h, maximum 49 h). Medium roast with coarse grinding degree was preferred. Following the brewing of the cold brew, the average storage time was one day (median 0.6 days, standard deviation 1.4 days, maximum seven days).

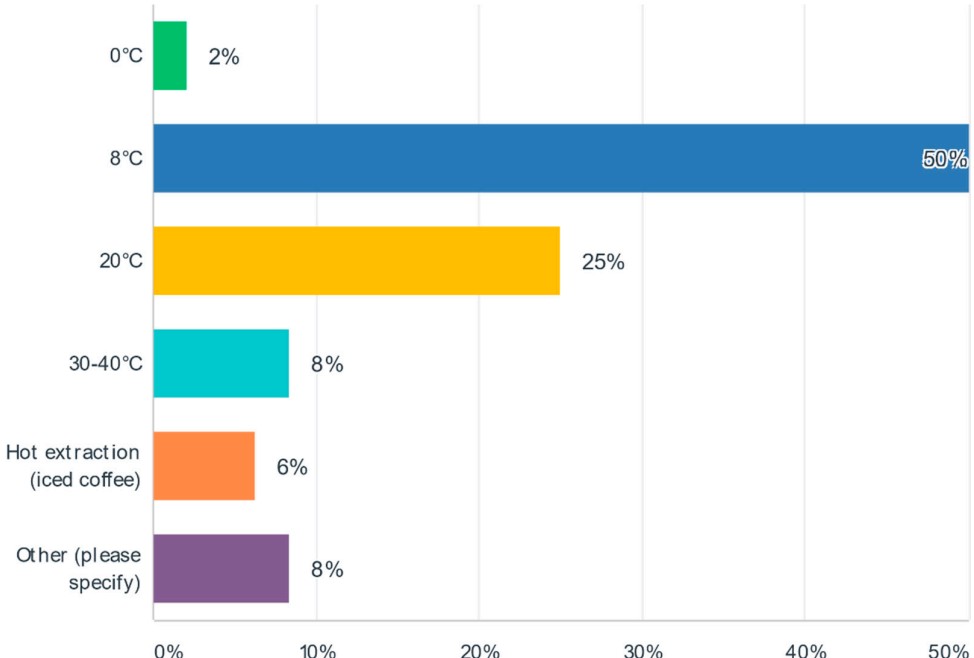

**Figure 7.** Initial survey result of cold brew extraction temperatures.

## 5. Open Problems and Challenges about Cold Brew Coffee

### 5.1. Should Cold Brew Be Served with Ice?

The ice will lower the temperature and may also dilute the aromas. Cold temperatures will also suppress the optimal taste perception. Therefore, a good cold brew does not need ice. However, obviously, if consumers like ice, such as in whisky-like cocktails, there is no reason not to offer it.

### 5.2. Is Cold Brew Extracted at Room Temperature Still a "Cold Brew"?

There is no available definition or regulatory standard on cold brew coffee. According to the authors' own judgement, every brew made at temperatures below body temperature may be considered as "cold" brew. Cold does not necessarily imply "ice cold". We, however, no longer consider hot brewed (>65 °C) coffees that are just cooled down, as cold brew. If cold brews are brewed at room temperature (which could reach levels above 30 °C in some regions of the world), it must obviously be considered that the extraction will be much faster. This could even be advantageous regarding microbiological contamination problems (e.g., comparing 18 h at 8 °C vs. 2 h at 20 °C). However, systematic research into this issue is lacking so far. Shelf life should be longer at fridge temperature than at room temperature.

### 5.3. What Would Your Suggestion Be for Cold Brew Extraction Conditions?

Based on the experience at Coffee Consulate, an amount of 80 g/L coffee is suggested for 2 h at initially 15 °C (tap water temperature in Europe) then stored in a fridge [19]. We use a 5 L food-grade plastic container with the aliquot of medium fine coffee (Catuai, pulped natural, or natural) and water (4.8 °dH German hardness), then we stir once after 1 h of lixiviation. Finally, the product is filtered using a standard paper filter. We believe that this method is simple and stable, making a consistent product as proven by various coffee shops that use this recipe.

From a sensorial standpoint, nothing may be gained by prolonging the extraction time to 4 h or even longer [19]. Increasing the extraction time may only increase the bitterness of the beverage. Some empirical evidence confirms this opinion: Caffeine and 3-chlorogenic acid reached equilibrium (i.e., 100% extraction, compare Figure 4) between 6 and 7 h, instead of 10 to 24 h outlined in some typical cold brew methods [25]. Cold drips were recognized as more bitter and temperature was found to increase the concentration of several compounds [13]. We are currently in the process of adjusting our coffee analytical methods [26–28] to cold brews aimed at studying and optimizing the multivariate influences and their interactions on cold extraction.

### 5.4. Is It Possible to Make Cold Brew with Beans from the Asia-Pacific Region?

These beans are often underrated. The advantage of the Indonesian and South-East Asian (e.g., Myanmar, Thailand, Philippines, and Malaysia) beans is the uniqueness in very floral and spicy coffees. To make a cold brew out of these beans really needs consideration of roast profile and extraction method. Cold brew prepared with beans from Indonesia (Arabica Bali Kintamani coffee) showed a fruity flavor with intense sweetness [29]. Otherwise, no reports on the sensorial data of South East Asian (SEA) coffees in cold brew are currently available.

### 5.5. What Would Be the Optimum Shelf Life of a Cold Brew Coffee?

Optimally, the cold brew would sell on the same day of preparation. The authors are of the strong belief that cold brew should be made fresh everyday (this actually appears to be in divergence with current practices, see survey result above). When stored for longer, some changes in product quality can be expected, such as raised acidity and greater ethanol content, also decreases in sugar content/sweetness due to yeast or bacteria activity. For example, So et al. [30] have shown that pH

declines and total acidity increases during cold brew storage for eight weeks, with greater effects for storage at 20 °C than at 4 °C.

In addition to microbiological changes, the product will also go rancid and stale due to an oxidative process. This oxidative process will not be as fast as for hot brewed coffee, which has a much higher lipid content. Anecdotally, if cold brew is stored under nitrogen pressure (e.g., in kegs), oxidation can be avoided and shelf-life extended [31,32]. Some studies claim that the shelf-life of cold brew is limited not by microbial stability, but rather by deterioration in sensory attributes [33].

Currently, for artisanal producers, we would suggest a maximum storage time of two days. Taste is currently the only guidance available on site to determine how long a cold brew may be stored. Otherwise, microbiological laboratory testing for shelf life analyses might be conducted.

To give an analogy with filter coffee, this product would turn stale and be thrown away after one to two hours. We do not believe that there is a reason to demand week-long storage times for cold brew, which basically has the same underlying costs (similar brewing ratio) as filter coffee. We know from experience that this also greatly increases customer satisfaction. It is a shame that the segment of cold brew, at least in Central Europe, appears currently dominated at an industrial level by "called brews" (i.e., hot extracted fakes) and on the artisanal level by over-extracted and over-stored products, which are sometimes extremely acidic and bitter, which is adverse to the goal of achieving a loyal, returning customer.

### 5.6. Is It Possible to Prepare Cold Brew as Concentrate and Dilute before Serving?

If a high-quality coffee is used to prepare a concentrate, there should be no reason that it should taste bad. This would be the equivalent of high gravity mashing on beer brewing. Increasing the ground coffee to water ratio may simplify the extraction and storage of the brew, that can be diluted just before consumption. Most commercially available concentrates, however, appear to be based on lower grade coffees and bad roasts as well as on hot extraction, using advanced equipment such as evaporation systems. Currently, we would not advise making concentrates in smaller coffee shops. There is also no research available on how concentration influences flavor.

### 5.7. What Are Typical Customer's Complaints Against Cold Brew?

It is too bitter, because bitterness is the sign that something is turning toxic. The next complaint would be too sour or too acidic (meaning that it tastes rotten). The acidity may arise from over-extraction and microbiological spoilage (lactic characteristics). These defects can be avoided when over-extraction and over-storage is avoided.

### 5.8. What Are the Risks of Handling Cold Brew Coffee?

Cold brew has a pH value greater than 4.6 (typically 4.9–6.0 [25,34,35]), which is a low acidity and does not effectively suppress microbial growth [5]. Therefore, foodborne pathogens (i.e., human illness causing microorganisms) or spoilage organisms, which mainly affect flavor, may develop in cold brew [5], see also Section 3.3. above.

Some initial experiments showed that the growth of pathogens may be inhibited by compounds in the coffee, but this must be confirmed for each individual preparation [5]. Conversely, spoilage organisms such as molds and yeasts appear not to be inhibited and may increase quickly during storage leading to fermentation [5]. Most sensitive products are ready-to-drink beverages, especially those under anaerobic conditions (for example when filled under pressure with nitrogen [34]). These products must be closely controlled to mitigate the risk of *Clostridium botulinum* (botulism), by either pH control or thermal processing [36]. In fact, the product "Death Wish" nitro cold brew was recalled from the market in September 2017, because it was determined that its production process could lead to the growth and production of botulinum toxin in this low-acid food commercialized in reduced oxygen packaging [37,38].

We would currently compare the risk of cold brew coffee to other alcohol-free beverages, for which considerably more experience exists. As cold-brew has a small amount of sugars (e.g., some initial findings found cold brew higher in reducing sugars than hot brewed coffee [39]), and unlike beer, does not contain alcohol which inhibits bacterial growth, the risk would be similar to, e.g., other alcohol-free beverages with pH > 5. Hence, special care and diligence need to be applied regarding handling and cleaning of dispensing equipment and coffee lines (e.g., for nitro cold brew "on tap"). For example, the German norm DIN 6650-6 suggests a cleaning and disinfecting interval of at least once a day for alcohol-free beverages [40].

Special care is also required when cold brew is filled in kegs needing careful study of the shelf life [6]. Most commercial cold brews in the market also require refrigeration for both storage and dispensing [41]. We recommend to specifically consider cold brew in the Hazard Analysis Critical Control Points (HACCP) system, which is mandatorily implemented in many countries including the European Union. To avoid contamination with pathogenic microorganisms, maximum storage times based on microbiological shelf-life testing should be implemented (including labelling of maximum duration and compliance controls that the product is removed before this date). Visual controls for clouding and organoleptic testing at least once a day before service starts should be conducted.

Some further guidelines are specified in the "Cold brew coffee toolkit for industry" published by the National Coffee Association of U.S.A. [36]. The toolkit includes information on challenge studies to avoid risk from spore-forming bacteria such as *Clostridium* and guidance on shelf-life testing, storage, and handling [36]. The British Columbia Centre for Disease Control suggests a maximum refrigerated storage of less than 10 days for products stored at 4 °C and below, if no other controls are present [34]. The Centre also suggests the following options to control the hazards of spore-forming bacteria for products intended for longer term storage: (i) Heating and pasteurization, (ii) reducing the pH to 4.6 or below, (iii) ensuring aseptic processing, (iv) addition of preservatives, and (v) a combination of these controls [34].

*5.9. What Are the Advantages of the Cold Drip Method?*

Based on the experience at Earthlings Coffee Workshop, customers like the concentrated, punchy flavor from ice drip coffee. It is usually served straight or with ice cubes, dependent on the customers' preference. The ratio is usually 1:5. 300 g of blend and about 1450 g of ice cubes plus 50 g of filtered water. The time is about 8–12 h. The use of ice cubes is extremely important for the flavor.

## 6. Conclusions

We are at the very beginning of scientifically understanding what cold brew coffee is. It is important not to have a pre-mindset into any direction and be really open to experience the entire spectrum of possibilities with this extraction technology. The cold brew extraction is a highly multivariate process [13], and additional chemical, microbiological, and sensory studies are needed to increase our understanding. Additionally, the taste profile of cold brew can be influenced by the serving, e.g., nitrogen infused or so-called nitro cold brew, milk addition, sugar addition, use of ice, or other ingredients such as alcoholic beverages. The nitro method is specifically interesting because it considerably changes the look and taste of the beverage. The nitrogen gas percolates through the glass like a Guinness beer, leaving a creamy head on the top [6]. The flavors are dispersed across the tongue because the bubbles dramatically increase the surface giving a smooth and creamy taste [6]. In our experience, the nitro method also intensifies the sweetness and takes the bitterness out of the product.

Finally, the food safety hazards of cold brew need increased consideration. As suggested by Lopez [15], both regulatory agencies as well as industry should address the issue aiming for standards and good manufacturing practices reducing the risks of cold brew coffee.

**Author Contributions:** Conceptualization, S.S. and D.W.L.; formal analysis, D.W.L.; investigation, S.S. and L.C.; methodology, S.S., L.C., and D.W.L.; project administration, D.W.L.; supervision, S.S. and D.W.L.; visualization,

Steffen Schwarz; writing—original draft, D.W.L.; writing—review and editing, R.K., K.L.W.T., S.S., and L.C. All authors have read and agreed to the published version of the manuscript.

**Funding:** This research received no external funding.

**Acknowledgments:** This article is based on training sessions by professional coffee training centers in Malaysia and Germany. The training sessions were initiated by presentations of invited speakers followed by Q&A sessions. All participants in the online training sessions are thanked for their attendance, questions and suggestions. Photogem25/Fiverr is thanked for proofreading the revised manuscript version.

**Conflicts of Interest:** The authors declare no conflict of interest.

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
