# Peer review of "Current Challenges of Cold Brew Coffee—Roasting, Extraction, Flavor Profile, Contamination, and Food Safety"

_challenges, doi:10.3390/challe11020026_

Round 1

Reviewer 1 Report

The topic of cold-brew is hot, and the authors' experience in this field can be of great interest or the readers. However, the structure of the paper is quite odd for a scientific journal.

Authors claim to not having much literature on this field (L183) but, although may be true in near past a quick search in Goggle Scholar (clearly not exhaustive) rends 3 papers published in 2019 and 7 published on 2020. A more accurate search could have provided more solid grounds for this review.

A short list of relevant papers (not included in the present review) are:

  • Angeloni, G.; Guerrini, L.; Masella, P.; Innocenti, M.; Bellumori, M.; Parenti, A. “Characterization and comparison of cold brew and cold drip coffee extraction methods.” J. Sci. Food Agric. 2019, 99, 391–399.
  • Contador, Marina, Flavio Schmidt, and Ana Braga. "Otimização do processo de extração de café a frio (cold brew coffee), caracterização físico-química e sensorial." Revista dos Trabalhos de Iniciação Científica da UNICAMP 27 (2019): 1-1.
  • Hamilton, Leah M., and Jacob Lahne. "Assessment of instructions on panelist cognitive framework and free sorting task results: A case study of cold brew coffee." Food Quality and Preference 83 (2020): 103889.
  • Han, Ji-Won, Hoon Boo, and Myung-Sub Chung. "Effects of extraction conditions on acrylamide/furan content, antioxidant activity, and sensory properties of cold brew coffee." Food Science and Biotechnology (2020): 1-10.
  • Rao, Niny Z., Megan Fuller, and Meghan D. Grim. "Physiochemical Characteristics of Hot and Cold Brew Coffee Chemistry: The Effects of Roast Level and Brewing Temperature on Compound Extraction." Foods 9.7 (2020): 902.

Another recently published paper can be relevant for the discussion, as compare different extraction temperatures (although addressed to espresso coffee (Foods 2020, 9(1), 36; https://doi.org/10.3390/foods9010036)

Authors insist on the risk of contamination of cold-brew coffee. In the Extraction degree chapter (L111) there is a link between extraction time and contamination. Although authors do not precise it, it can be assumed they refer to microbial contamination. What contamination can occur during coffee brewing (extraction)? If authors refer to cold brewing, should organise better the discussion distinguishing the different brewing Systems.

On the paragraph L123-128 authors compare different extraction yields. How was the 100% yield stablished? Then, authors good and bad flavours, indicating different kinetics on flavour compounds extractions. This should be described in a more rigorous way, indicating the solubility of the different substances. Or at least indicating which compounds can contribute favourably to the brewed coffee aroma and which ones are distasteful.

Paragraph L141-147 about serving systems is not in line with the paper, that deals with extraction temperature rather than with on the confounding variables. I suggest to delete it.

Figure 4 cannot be read in my pdf version. Ensure a high-resolution version to be published. Provide a proper reference for the figure.

On the sensory description of brews, have the authors applied a statistical test to their results? If so, report the test applied and the degree of significance. If not, moderate the assertiveness and avoid the use of “evident” or “clearly”.

The link on Figure 7 is no longer valid, and can be deleted.

I strongly recommend the authors to continue their investigations in cold brew. They clearly have a lot of knowledge and are asking the right questions. However, a scientific publication requires a stronger fact-based discussion for which analytical data would be required.

Author Response

The topic of cold-brew is hot, and the authors' experience in this field can be of great interest or the readers. However, the structure of the paper is quite odd for a scientific journal.

Authors claim to not having much literature on this field (L183) but, although may be true in near past a quick search in Goggle Scholar (clearly not exhaustive) rends 3 papers published in 2019 and 7 published on 2020. A more accurate search could have provided more solid grounds for this review.

Thank you for the assessment of our paper. The major intention of our paper was to point out the research needs and not to provide a comprehensive review. Actually, we did comprehensive searches in July 2020 in PubMed, Web-of-Science, Food Science and Technology Abstracts, Google Scholar and Research Gate. Based on these searches, we included some important references, but excluded some references that were not relevant to our questions.

A short list of relevant papers (not included in the present review) are:

  • Angeloni, G.; Guerrini, L.; Masella, P.; Innocenti, M.; Bellumori, M.; Parenti, A. “Characterization and comparison of cold brew and cold drip coffee extraction methods.” Sci. Food Agric. 2019, 99, 391–399.
  • Contador, Marina, Flavio Schmidt, and Ana Braga. "Otimização do processo de extração de café a frio (cold brew coffee), caracterização físico-química e sensorial." Revista dos Trabalhos de Iniciação Científica da UNICAMP 27 (2019): 1-1.
  • Hamilton, Leah M., and Jacob Lahne. "Assessment of instructions on panelist cognitive framework and free sorting task results: A case study of cold brew coffee." Food Quality and Preference 83 (2020): 103889.
  • Han, Ji-Won, Hoon Boo, and Myung-Sub Chung. "Effects of extraction conditions on acrylamide/furan content, antioxidant activity, and sensory properties of cold brew coffee." Food Science and Biotechnology (2020): 1-10.
  • Rao, Niny Z., Megan Fuller, and Meghan D. Grim. "Physiochemical Characteristics of Hot and Cold Brew Coffee Chemistry: The Effects of Roast Level and Brewing Temperature on Compound Extraction." Foods 9.7 (2020): 902.

Thank you for pointing out these papers, some of which were published during the peer review process. We have included the relevant references.

Another recently published paper can be relevant for the discussion, as compare different extraction temperatures (although addressed to espresso coffee (Foods 2020, 9(1), 36; https://doi.org/10.3390/foods9010036)

Thank you for pointing out this reference. We have included it when we describe the similarity of espresso. It would be interesting to replicate this research including lower extraction temperatures.

Authors insist on the risk of contamination of cold-brew coffee. In the Extraction degree chapter (L111) there is a link between extraction time and contamination. Although authors do not precise it, it can be assumed they refer to microbial contamination. What contamination can occur during coffee brewing (extraction)? If authors refer to cold brewing, should organise better the discussion distinguishing the different brewing Systems.

There is clearly a hazard of microbiological contamination whenever coffee is extracted with cold water independent of the brewing system for cold brew (e.g. see the NCA industry toolkit). We have revised the text throughout to clarify the nature of contamination.

On the paragraph L123-128 authors compare different extraction yields. How was the 100% yield stablished? Then, authors good and bad flavours, indicating different kinetics on flavour compounds extractions. This should be described in a more rigorous way, indicating the solubility of the different substances. Or at least indicating which compounds can contribute favourably to the brewed coffee aroma and which ones are distasteful.

Our definition of extraction degree is explained in the first sentence of section 3.3.

Paragraph L141-147 about serving systems is not in line with the paper, that deals with extraction temperature rather than with on the confounding variables. I suggest to delete it.

To increase clarity, we have decided to move the information on serving systems from section 3.4 to section 6.

Figure 4 cannot be read in my pdf version. Ensure a high-resolution version to be published. Provide a proper reference for the figure.

Thank you! We have exchanged the figure.

On the sensory description of brews, have the authors applied a statistical test to their results? If so, report the test applied and the degree of significance. If not, moderate the assertiveness and avoid the use of “evident” or “clearly”.

The terms were toned down as requested.

The link on Figure 7 is no longer valid, and can be deleted.

Thank you. The link became invalid during peer review and was deleted.

I strongly recommend the authors to continue their investigations in cold brew. They clearly have a lot of knowledge and are asking the right questions. However, a scientific publication requires a stronger fact-based discussion for which analytical data would be required.

This is actually the reason why we chose the “Challenges” journal for this article. The scope reads: “it publishes scholarly content which is typically not publishable in traditional research journals, such as research proposals (funded and unfunded), registered reports (study protocols prior to experiments being conducted), research plans (e.g., a protocol for a systematic review, further research of a technology application, etc.), research or technology ideas, policy studies relating to science and scholarly research, open contests aimed at solving grand challenges, prize announcements, description of prototypes, calls for, or description of, international research collaborations or complementary support, etc.” We have changed the article type to “viewpoint” to better state to the reader that this is not a traditional review article.

Reviewer 2 Report

The authors bring up a relevant topic on which little research is conducted so far. Unfortunately, the authors do not take the opportunity to present any relevant and solid new research. The paper does not bring any scientific relevance as it is a blend of review papers, experience, but mostly the paper describes the authors opinion without a scientific base.

Author Response

Thank you for evaluating our paper. According to the editorial suggestion, we have changed the article type to better show that this is not a classical research or review article. Please note the scope of the journal Challenges (see last response to reviewer #1 above). We believe that opinion papers that point our research needs also should have a place in scientific publications.

Reviewer 3 Report

Generally, the review is of great interesting, the topic here is very attractive. It’s true that more scientifically understanding should be started for these new type of beverage. However, the statement is too colloquial, and more scientific description should be used.

An example is shown here, such as for section 3.2.

Line 88~100  The section title is ‘Hazards during cold brew extraction’, a whole reported list of all the possible hazards should be included, not just a general description.

  1. Line 91, the author mentioned that ‘with cold brew there are a lot of hazards’, for a lot of harzards, what are they? Based on the current information, only organisms had been mentioned, what about the chemical contaminants? Such as pesticide, mycotoxin, these chemical contaminants are import part of hazards. And there are also publication data, the author should discuss them as well. ‘A lot of’ is not just organisms.
  2. And on the other hands, are they safe enough to drink? Is there any reported data? What about the reported hazards, permitted intake level?

More experiment data, special those data published in journal like Food Chem, LWT, ect., should be used as references.

Author Response

Generally, the review is of great interesting, the topic here is very attractive. It’s true that more scientifically understanding should be started for these new type of beverage. However, the statement is too colloquial, and more scientific description should be used.

The text was revised throughout to avoid colloquialisms. We also have included further references to back up the science (see reviewer #1).

An example is shown here, such as for section 3.2.

Line 88~100  The section title is ‘Hazards during cold brew extraction’, a whole reported list of all the possible hazards should be included, not just a general description.

The section about contaminations in 3.2 was expanded and clarified.

Line 91, the author mentioned that ‘with cold brew there are a lot of hazards’, for a lot of harzards, what are they? Based on the current information, only organisms had been mentioned, what about the chemical contaminants? Such as pesticide, mycotoxin, these chemical contaminants are import part of hazards. And there are also publication data, the author should discuss them as well. ‘A lot of’ is not just organisms.

The sentence was changed and information on the other chemical contaminants was added. However, we strongly believe that the difference between cold brew and hot brew in hazard profile is clearly microbiological contamination. There should not be a difference in water soluble chemical contaminants such as acrylamide.

And on the other hands, are they safe enough to drink? Is there any reported data? What about the reported hazards, permitted intake level?

Basically, the product is safe if typical hygiene rules are followed (such as the suggestions of the NCA industry toolkit). However, we were unable to identify systematic monitoring data on cold brew contamination.

More experiment data, special those data published in journal like Food Chem, LWT, ect., should be used as references.

See response to reviewer 1. We have included additional papers as requested.

Reviewer 4 Report

The authors don’t include a large number of scientific papers published in recent times about this argument. In my opinion, the aim of this research was properly trying to collect several research papers and try to evaluate future challenges.    

I think is not appropriate to Indicate the materials and methods, in fact, the section is empty, and is indicated only the words selected for the research (very poor).

Table 1 is not helpful, is important underline the different from cold extraction and a normal extraction

Line 70-75, in the scientific literature, are presents several articles that comparing cold different extraction methods, cold brew, and cold drip for example.

Line 108, which is the author's experience? is not explained. 

The authors should describe the concept of turbulence because in this manner it seems a confusing concept.

Figure 4 is not clear. It’s impossible to understand.

Is not clear why the authors don’t follow a specific order to evaluate the parameters. It could probably more clear if the authors follow the process order to produce the cold brew, starting to the roasting process, then grinding step followed by the extraction. All this part, regarding the process, is not explained clearly and is not exhaustive and requires a deeper study about the scientific literature. 

Is probably more interesting and better explained the second part regarding future challenges.

Also, the conclusions do not provide any information and could be improved.

Author Response

The authors don’t include a large number of scientific papers published in recent times about this argument. In my opinion, the aim of this research was properly trying to collect several research papers and try to evaluate future challenges.    

As mentioned in response to reviewer #1 above, we have done searches in several databases and included the most relevant references. As requested also by the other reviewers, we have expanded the reference list. However, it must be mentioned that this article was not intended as systematic review.

I think is not appropriate to Indicate the materials and methods, in fact, the section is empty, and is indicated only the words selected for the research (very poor).

Sorry, but the materials and methods section just specifies what we did in a concise fashion. We could delete it, but we believe that it is more appropriate to leave the section as is.

Table 1 is not helpful, is important underline the different from cold extraction and a normal extraction

The table was deleted as requested. The information is stated in the text.

Line 70-75, in the scientific literature, are presents several articles that comparing cold different extraction methods, cold brew, and cold drip for example.

Several references were added to the section in lines 70-75 regarding the different cold brew extraction methods.

Line 108, which is the author's experience? is not explained. 

Details about our experience were added.

The authors should describe the concept of turbulence because in this manner it seems a confusing concept.

The concept of turbulence is explained in line 83: increasing the water contact into the coffee grounds by stirring. Typically, during cold brew extraction, regular stirring is recommended (see, e.g. recommendations by commercial systems such as Toddy).

Figure 4 is not clear. It’s impossible to understand.

Figure 4 has been updated.

Is not clear why the authors don’t follow a specific order to evaluate the parameters. It could probably more clear if the authors follow the process order to produce the cold brew, starting to the roasting process, then grinding step followed by the extraction. All this part, regarding the process, is not explained clearly and is not exhaustive and requires a deeper study about the scientific literature. 

The flow of the section was revised to start with the roasting process followed by the extraction process.

Is probably more interesting and better explained the second part regarding future challenges.

Thank you.

Also, the conclusions do not provide any information and could be improved.

The conclusion was expanded by re-arranging some of the material that better fits to this section.

Round 2

Reviewer 1 Report

I consider the authors have respond to the previous questions, and incorporate most of the suggested changes. The article is accpetable in this present form

Reviewer 4 Report

The authors have followed the advice from different reviewers and have modified and improved the article. 
In this form, the article is correct and suitable for publication.

This manuscript is a resubmission of an earlier submission. The following is a list of the peer review reports and author responses from that submission.

Round 1

Reviewer 1 Report

The paper corresponds better to a review rather than to a research paper. This section 2 on Materials and Methods is not relevant and I suggest to be deleted.

L77-82 Grinding factor is repeated as the first item on the list (L78), again on L79 and repeated as the last sentence on the paragraph (L81). Organise jointly the three instances for this relevant factor for coffee brewing.

L89. Accepting that microbial growth on cold brew coffee is possible, it should not be focused only on spoilage but also on the risk of pathogenic microorganisms. Although being discussed in a later part of the document, the analysis of the risk of pathogens like Listeria or Salmonella should not be avoided here.

L95. Pasteurisation and sterilisation are not interchangable concepts.

L113. Authors suggest that low-temperature brewing is a riskier practice because of long infusion times. The risk of microbial growth is strongly related to the extraction temperature and is dramatically reduced at really cold temperatures.

L132-136. This nitro method, as described, is no a brewing system but just a serving method and thus more similar to iced coffee than the cold brew system discussed in the paper. I suggest to remove it.

L162-163. It is not relevant to refer the work as to a BSc thesis. The sentence "In the context of research for the bachelor thesis of L.C.," can be just removed.

L172-173. It is not relevant to refer the work as to a BSc thesis. The sentence "within the context of the bachelor thesis of L.C." can be just removed.

L252-257. In this Q&A section, this question 5.6 can refer to the equivalent to high gravity mashing on beer brewing. Increasing the ground coffee to water ratio simplifies the extraction and storage of the brew, that can be diluted jut before consumption. The concentration of the brew by evaporation is another business.

L281. Lemonades or cola beverages are cot comparable to coffee, as the pH is really low in these soda drinks

L296. Clostridium should appear in italics.

Author Response

The paper corresponds better to a review rather than to a research paper.

Thank you!

This section 2 on Materials and Methods is not relevant and I suggest to be deleted.

We believe that review papers may also have a methods section, specifying search strategy, databases, inclusion and exclusion criteria etc.

L77-82 Grinding factor is repeated as the first item on the list (L78), again on L79 and repeated as the last sentence on the paragraph (L81). Organise jointly the three instances for this relevant factor for coffee brewing.

The double mentioning of grinding was deleted, thank you. On line 81, grinding is mentioned for a second time on intention to specify its importance. We have revised the section to improve understanding.

L89. Accepting that microbial growth on cold brew coffee is possible, it should not be focused only on spoilage but also on the risk of pathogenic microorganisms. Although being discussed in a later part of the document, the analysis of the risk of pathogens like Listeria or Salmonella should not be avoided here.

Thank you! “Spoilage” was changed to “contamination” to make it more general. The problem of pathogenic microorganisms was included as suggested.

L95. Pasteurisation and sterilisation are not interchangable concepts.

Thank you. “Such as“ was changed to “or” to specify that these are different processes.

L113. Authors suggest that low-temperature brewing is a riskier practice because of long infusion times. The risk of microbial growth is strongly related to the extraction temperature and is dramatically reduced at really cold temperatures.

Thank you. We have included this point in line 90.

L132-136. This nitro method, as described, is not a brewing system but just a serving method and thus more similar to iced coffee than the cold brew system discussed in the paper. I suggest to remove it.

Cold-brew is often served using the nitro method, which considerably influences its taste. We might go so far to say that cold brews from very low-quality coffees, which are typically over-roasted and over-extracted, only become drinkable as nitro cold brew. Therefore, the authors decided to retain the remarks about nitro.

L162-163. It is not relevant to refer the work as to a BSc thesis. The sentence "In the context of research for the bachelor thesis of L.C.," can be just removed.

Deleted as requested.

L172-173. It is not relevant to refer the work as to a BSc thesis. The sentence "within the context of the bachelor thesis of L.C." can be just removed.

Deleted as requested.

L252-257. In this Q&A section, this question 5.6 can refer to the equivalent to high gravity mashing on beer brewing. Increasing the ground coffee to water ratio simplifies the extraction and storage of the brew, that can be diluted just before consumption. The concentration of the brew by evaporation is another business.

Thank you for providing the example of high-gravity brewing. We have revised this section accordingly and included these points.

L281. Lemonades or cola beverages are not comparable to coffee, as the pH is really low in these soda drinks

Thank you. Lemonades and cola beverages was deleted.

L296. Clostridium should appear in italics.

Corrected. Thanks!

Reviewer 2 Report

General comments on the quality of the research presented:

The authors have chosen a very important and relevant topic of research. Indeed only little data are available in literature on cold brew and research is required to get a better understanding of this beverage. Two topics are treated: 1) extraction and in cup result and 2) food safety. The two topics are very different and would merit a paper dedicated to each.

Data presented in the paper to a large extent based on authors opinion rather than a factual base (examples L 39, 198, 214, 252). More research data would be required to substantiate the opinions. According to the journal aims: For research and development articles, full experimental details must be provided so that the results can be reproduced. 

Food safety is discussed, but no analytical data on contaminations are presented. I therefore wonder what brings the authors to provide their opinions in a scientific journal.

Sensorial preferences and sensorial profiles are described. Please provide the methodologies used and the number of consumers / tasters.

The Discussion is stated as a question and answer. New topics are brought into the discussion that were not researched in the paper. It makes me wonder why this format is chosen? Who asks the questions? Why has the more recognised format of a discussion (where the research data presented are discussed in a wider context) not been applied?

Detailed comments:

Table 1 : Iced coffee. On what assumption is the ‘emphasises oil soluble aromas’ based? Is it not the same as the aromas in hot coffee, i.e. aromas having a broad range of polarities? Would it not be more correct to speak of a balance towards more polar compounds vs a balance where the less polar compounds will increase?

Figure 2: It is not clear if the authors refer to the time and temperature of extraction (due to the fact that other parameters such as the turbulence clearly are extraction related) or to time and temperature of storage. The latter would probably have a much larger impact on contamination. Furthermore, parameters such as sterilizing materials, closed containers and other good manufacturing process parameters should be mentioned. Please also explain how turbulence, water hardness and strength would impact contamination. It is not clear.

Yield: The authors speak of an optimum of 70% yield and refer to extracting less than 100% yield in specialty coffee. Please define yield and make sure it is understood that you are not using the same definition as the commonly used 'yield' in the coffee brewing chart (optimum as defined from 18-22%).

Figure 5 : It should be clarified at which temperature both coffees were tasted. If the temperature was not the same for both, this has to be clearly indicated as a contributing effect. It is known that when a coffee cools down, fruity and floral notes change and become more prominent. Please also explain methodology and number of tasters.

Figure 6 / L 150: Optimised roasting profiles are presented. It is however not clear on what scientific base ‘optimised’ is based. Data supporting the statement are missing.

L 178: The authors should not place a surveymonkey link as invite for readers to take the survey, and a 2nd link to review the results.  I suggest removing the links and placing a summary of the questions brought forward in the survey so that people reading the paper after July 2020 also will understand what the survey was about.

Figure 7: This brings no additional value to the reader as the information is already fully understood from the text.

L 198: The authors answer a question by ‘we consider’ but do not explain on what facts they base their opinion. Furthermore, they speak about the risk of contamination without giving any data on e.g. microbiological growth.

L 214: The authors speak of their experience, but why are the data not presented? The topic of extraction time is clearly a very relevant and interesting one as one of the hurdles in cold brew is the time of extraction. Presenting sensorial data of how the cold brew changes with extraction time, or with temperature would be highly relevant for the industry. A model using water temperature vs extraction time as a function of yield and sensory attributes could be a way to give the needed scientific fact based substantiation that this paper requires in order to be published.

L 221: Please present data on sensorial properties of Asian coffees in cold brew. Otherwise, please remove.

L 227: The question asked seems to be answered by an opinion. I have however not seen any scientific evidence that supports the opinion. Speaking of contamination risk mandates data (not a recommendation that analysis can be done), and the appearance of staling would be expected to be supported by sensorial data.

L 252: Opinion based

L 259: Bitterness is a sensorial feature in coffee. Overextraction increases bitterness. Why do the authors state that bitterness is a feature of toxicity? It seems very far fetched and, again, with no scientific substantiation. Similarly, sour may be perceived due to very light roast degree, not because the coffee is rotting.

Conclusion: I strongly recommend the authors to continue their investigations in cold brew. They clearly have a lot of knowledge and are asking the right questions. However, a scientific publication requires a stronger fact based discussion for which analytical data would be required.

Author Response

General comments on the quality of the research presented:

The authors have chosen a very important and relevant topic of research. Indeed only little data are available in literature on cold brew and research is required to get a better understanding of this beverage. Two topics are treated: 1) extraction and in cup result and 2) food safety. The two topics are very different and would merit a paper dedicated to each.

Thank you for your kind assessment and agreement on the importance of the topic. As there are only around 25 relevant references on both topics in the literature, we believe that this well fits into a single paper. In our pre-submission enquiries, the editor of “Beverages” advised us that review articles must contain more than 4000 words. As the article currently contains around 5000 words, we actually cannot see how this could be split into two articles.

Data presented in the paper to a large extent based on authors opinion rather than a factual base (examples L 39, 198, 214, 252). More research data would be required to substantiate the opinions. According to the journal aims: For research and development articles, full experimental details must be provided so that the results can be reproduced.

It is true that some of the arguments presented in this article are based on the considerable experience of the authors with cold brew, but in most instances backup from the scientific literature was possible, even if we agree that the evidence base is far from complete. See also the conclusion of the article. We have carefully looked though the text and toned down all arguments (e.g. examples in lines specified above) to clarify these points to the reader. As this is a review article and not an “research and development article”, we hope that the reviewer might advise discretion regarding the points above, considering that the article is for the Special Issue "Current Reviews in Beverages - 2021", which specifies the following aims  in the information for authors: “this Special Issue will provide an overview of these cutting-edge trends by promoting a multi-disciplinary perspective” and “new trends in beverage quality and safety control”. It must be considered that new and cutting-edge trends such as cold brew do not have similar databases to back up arguments, as if – for example – we had written a review about Pilsener beer.

Food safety is discussed, but no analytical data on contaminations are presented. I therefore wonder what brings the authors to provide their opinions in a scientific journal.

Actually, we presented some data (e.g. references 4,7 with some results on microbiological and chemical contaminants) as well as reference 21 with an industry toolkit to avoid contamination. Therefore, we deeply believe in our arguments and also know that our results are corroborated by unpublished industry data. All in all, we believe that the few available data along with the large interest in coffee industry about cold brew well justify the publication of the paper, which hopefully will stimulate much research on this topic on the future.

Sensorial preferences and sensorial profiles are described. Please provide the methodologies used and the number of consumers / tasters.

The methodology used for describing the sensorial profiles of the coffees has been done by Coffee Consulate flavour profile, which in contrast to other used coffee cup tasting methodologies does not rank by preference or give overall score but simply describes intensities of aromas (in eleven groups, which are divided in subgroups and freely associated „descriptors“), taste (sweet, salty, sour and bitter), since umami is a very uncommon taste in coffee and thus for simplification is left out, as well as haptic (body, mouthfeel and aftertaste).

All of the single parameters are described in an intensity level from 0 (lowest) to 4 (highest), while the aromas are described all only in 0,5 steps [i.e. 2,5], while taste and haptic are described in 0,1 steps [i.e. 2,6].

The document can be downloaded on the homepage from Coffee Consulate in the download-section freely (https://coffee-consulate.com/en/wp-content/uploads/2019/04/cc-flavour-profile_english-1.pdf) in several languages.

All cuptasters at Coffee Consulate are regularly cuptasting and calibrated in different panels. A panel consists out of a minimum of three or a maximum of 7 cuptasters.

The information and a link to the profile were implemented in the text.

The Discussion is stated as a question and answer. New topics are brought into the discussion that were not researched in the paper. It makes me wonder why this format is chosen? Who asks the questions? Why has the more recognised format of a discussion (where the research data presented are discussed in a wider context) not been applied?

As stated in the materials and methods section: “The major results of the review were discussed using several online training sessions with coffee experts hosted by the professional training centers of Earthlings Coffee Workshop (Malaysia) and Coffee Consulate (Germany). The feedback of the experts was included in the discussion section.” For clarification, we have changed the title of the discussion section.

Detailed comments:

Table 1 : Iced coffee. On what assumption is the ‘emphasises oil soluble aromas’ based? Is it not the same as the aromas in hot coffee, i.e. aromas having a broad range of polarities? Would it not be more correct to speak of a balance towards more polar compounds vs a balance where the less polar compounds will increase?

The changes were implemented as requested.

Figure 2: It is not clear if the authors refer to the time and temperature of extraction (due to the fact that other parameters such as the turbulence clearly are extraction related) or to time and temperature of storage. The latter would probably have a much larger impact on contamination. Furthermore, parameters such as sterilizing materials, closed containers and other good manufacturing process parameters should be mentioned. Please also explain how turbulence, water hardness and strength would impact contamination. It is not clear.

In Figure 2, only the extraction influences were described. The legend of Figure 2 was clarified.

Yield: The authors speak of an optimum of 70% yield and refer to extracting less than 100% yield in specialty coffee. Please define yield and make sure it is understood that you are not using the same definition as the commonly used 'yield' in the coffee brewing chart (optimum as defined from 18-22%).

The reviewer is correct that the word “yield” may be used not unambiguously in the coffee field. Therefore, we changed the word “yield” on all instances for better clarity.

Figure 5 : It should be clarified at which temperature both coffees were tasted. If the temperature was not the same for both, this has to be clearly indicated as a contributing effect. It is known that when a coffee cools down, fruity and floral notes change and become more prominent. Please also explain methodology and number of tasters.

The requested information was added to the text.

Figure 6 / L 150: Optimised roasting profiles are presented. It is however not clear on what scientific base ‘optimised’ is based. Data supporting the statement are missing.

The requested information was added to the text.

L 178: The authors should not place a surveymonkey link as invite for readers to take the survey, and a 2nd link to review the results.  I suggest removing the links and placing a summary of the questions brought forward in the survey so that people reading the paper after July 2020 also will understand what the survey was about.

The link to SurveyMonkey was deleted and a reference with a link to the full text of the questionnaire was added instead (see: https://www.researchgate.net/publication/340739045_Cold-Brew_Coffee_Preparation_Survey).

Figure 7: This brings no additional value to the reader as the information is already fully understood from the text.

Not all data shown in Figure 7 are stated in the text.

L 198: The authors answer a question by ‘we consider’ but do not explain on what facts they base their opinion. Furthermore, they speak about the risk of contamination without giving any data on e.g. microbiological growth.

The definition of cold brew around line 198 was clarified. The authors have often observed spoilage of cold brew worldwide, so there clearly is a risk, even if the literature base is weak. The microbiological risk is also acknowledged by industry (see: Ref 21).

L 214: The authors speak of their experience, but why are the data not presented? The topic of extraction time is clearly a very relevant and interesting one as one of the hurdles in cold brew is the time of extraction. Presenting sensorial data of how the cold brew changes with extraction time, or with temperature would be highly relevant for the industry. A model using water temperature vs extraction time as a function of yield and sensory attributes could be a way to give the needed scientific fact-based substantiation that this paper requires in order to be published.

The scope of the paper was to review the literature rather than to include own investigations. We agree with all points of the reviewer and will conduct and publish such research in the near future. However, as specified in the paragraph, there is even empirical evidence available in the literature which confirms our experience. We have revised the text to make the points clearer.

L 221: Please present data on sensorial properties of Asian coffees in cold brew. Otherwise, please remove.

There are no written reports on sensorial data of South East Asian (SEA) coffees in cold brew apart from the cited reference. The general notes of those coffees already described in books of the green coffee trade summarize the coffees from SEA as „floral, spicy and fruit dominated“ (The coffee companion - A connoisseur’s guide, Jon Thorn). It is mainly by experience of the cuptasters of Coffee Consulate, that have cup tasted coffees over 15 years from all over the world in all styles of preparations, that have given the indication to answering the question in this way.

L 227: The question asked seems to be answered by an opinion. I have however not seen any scientific evidence that supports the opinion. Speaking of contamination risk mandates data (not a recommendation that analysis can be done), and the appearance of staling would be expected to be supported by sensorial data.

There are several references presented to confirm our “opinion”: For example, So et al. [15] have shown that pH declines and total acidity increases during cold brew storage for 8 weeks, with larger effects for storage at 20°C than at 4°C. In addition to microbiological changes, the product will also go rancid and stale due to oxidative processes. This oxidative process will not be as fast as for hot brewed coffee, which has much higher lipid content. Anecdotally, if the cold brew is stored under nitrogen pressure (e.g. in kegs), oxidation may be avoided and shelf life be extended [16,17]. Some studies claim that the shelf life of cold brew is limited not by microbial stability, but rather by deterioration in sensory attributes [18].

L 252: Opinion based

The paragraph was revised, see also response to reviewer #1.

L 259: Bitterness is a sensorial feature in coffee. Overextraction increases bitterness. Why do the authors state that bitterness is a feature of toxicity? It seems very far-fetched and, again, with no scientific substantiation. Similarly, sour may be perceived due to very light roast degree, not because the coffee is rotting.

The paragraph with these points was deleted.

Conclusion: I strongly recommend the authors to continue their investigations in cold brew. They clearly have a lot of knowledge and are asking the right questions. However, a scientific publication requires a stronger fact-based discussion for which analytical data would be required.

Thank you for your conclusion and suggestions. The authors will follow up with research on cold brew.

Reviewer 3 Report

The authors reviewed cold brew coffee in terms of roasting, extraction, flavor profile, contamination, and food safety. This review article is poorly organized due to its low-quality structure. Overall, this manuscript is not acceptable as a review article.

The structure of this review article is not acceptable, which is more like a research article instead of a review article. It should be divided into subsections of roasting, extraction, flavor profile, contamination, and food safety to discuss each aspect one by one.

Line 193-308: The discussion is more like a Q&A section, which is not proper for the scientific writing of a review article.

The conclusions is relatively short for a review article.

Author Response

The authors reviewed cold brew coffee in terms of roasting, extraction, flavor profile, contamination, and food safety. This review article is poorly organized due to its low-quality structure. Overall, this manuscript is not acceptable as a review article.

The structure of this review article is not acceptable, which is more like a research article instead of a review article. It should be divided into subsections of roasting, extraction, flavor profile, contamination, and food safety to discuss each aspect one by one.

Thank you for reviewing our article. As requested, we have implemented subsections in part 3, which was rather long indeed.

Line 193-308: The discussion is more like a Q&A section, which is not proper for the scientific writing of a review article.

The discussion section was revised and renamed, also see comment to reviewer #1 above.

The conclusions is relatively short for a review article.

According to the journal’s guideline, the conclusion section is not mandatory, and we do not want to be very repetitious.

Round 2

Reviewer 2 Report

My comments from the first round are not fully answered nor is the paper significantly reworked to increase the scientific quality.

Reviewer 3 Report

N/A